# Fully Distributed Control for a Class of Uncertain Multi-Agent Systems with a Directed Topology and Unknown State-Dependent Control Coefficients

Zongcheng Liu [1],* , Hanqiao Huang [2],*, Sheng Luo [2], Wenxing Fu [2] and Qiuni Li [1],*

[1]  Aeronautics Engineering College, Air Force Engineering University, Xi'an 710038, China
[2]  Unmanned System Research Institute, Northwestern Polytechnical University, Xi'an 710072, China; luosheng611@163.com (S.L.); Wenxingfu@nwpu.edu.cn (W.F.)
*   Correspondence: liu434853780@163.com (Z.L.); cnxahhq@126.com (H.H.); lqnjk1@126.com (Q.L.); Tel.: +86-029-84787345 (Z.L.)

**Abstract:** To address the control of uncertain multi-agent systems (MAS) with completely unknown system nonlinearities and unknown control coefficients, a global consensus method is proposed by constructing novel filters and barrier function-based distributed controllers. The main contributions are as follows. Firstly, a novel two-order filter is designed for each agent to produce informational estimates from the leader, such that a connectivity matrix is not used in the controller's design, solving the difficultly caused by the time-varying control coefficients in a MAS with a directed graph. Secondly, combined with the novel filters, barrier functions are used to construct the distributed controller to deal with the completely unknown system nonlinearities, resulting in the global consensus of the MAS. Finally, it is rigorously proved that the consensus of the MAS is achieved while guaranteeing the prescribed tracking-error performance. Two examples are given to verify the effectiveness of the proposed method, in which the simulation results demonstrate the claims.

**Keywords:** distributed control; MAS; flight control



## 1. Introduction

The control of uncertain nonlinear systems has been researched for several decades, such that so many remarkable results have been obtained on this topic [1–9]. However, most of them are for SISO or MIMO systems, and their methods or techniques cannot be directly applying to multi-agent systems, as the information of each agent or subsystems is only available for part of others. According to the topology of information transformation graph, the graph can be divided into undirected and directed graphs. Generally, the consensus control of a MAS with the directed graph is more difficult than the undirected case, since the methods for the directed case are always applicable for the undirected case, but not vice versa.

Recently, some significant progress has been made in the control of a MAS [10–12]. For a linear MAS with undirect graphs, fully distributed adaptive consensus controller is present in [10]. Adaptive asymptotically consensus for an uncertain MAS is achieved in [11], and adaptive asymptotically consensus is achieved in [12] for an uncertain MAS, and so on. However, their methods are only applicable for a MAS with an undirected graph and are in vain for a MAS with a directed graph. For a MAS with a directed graph and constant control coefficients, adaptive consensus for a MAS with system nonlinearities satisfying match conditions is researched in [13] to solve the problem of actuator faults; a fully distributed adaptive consensus control is studied for a MAS with unknown control directions in [14] by using a Nussbaum gain technique; actuator faults in a MAS are considered in [15] with integral chain dynamics; and prescribed performance consensus control for uncertain MAS is investigated in [16]. Though much progress has been made [17–20], it should be noted

that there are still some nonnegligible problems to be solved. Firstly, the existing methods require the control coefficients to be constants, or even known, for a MAS with a directed graph. The main difficulty is that the Laplace matrix for a directed graph is asymmetric and thus the selections of control parameters must always resort to adaptive methods, which falls into trouble when the control coefficients are time-varying and unknown. Secondly, to the best of our knowledge, there is no global consensus control method for a MAS with a directed graph and the systems functions thereof completely unknown, except for [21], wherein the unknown system nonlinearities required to satisfy the Lipschitz conditions and control coefficients are one. Universal approximators such as neural networks (NN) or fuzzy logic systems (FLS) have been attempted to solve the consensus control problem of a MAS with completely unknown system nonlinearities [22–24], however, it is well known that these methods are semi-global in the sense that their stabilities depend on the initial conditions of systems and the careful selection of controller parameters. Therefore, NN or FLS-based approaches cannot guarantee the global consensus of the MAS, though they are very favorable to solve the problem of MAS with unknown nonlinearities.

As for the global control of systems with completely unknown nonlinearities, a pioneering work is [25], wherein a low-complexity controller is presented that cannot only achieve global convergence of all the system signals, but which can also guarantee the prescribed performance of tracking error and state errors. In view of the low complexity and strong robustness of this method, much research has been carried on this method for solving different nonlinear control problems [26–30]. By introducing a novel barrier function, a fault-tolerant controller is designed for a class of unknown nonlinear systems in [26]. With consideration to the constraints of system states, a barrier function-based adaptive control method is proposed in [27]. Addressing systems with unknown control direction and system dynamics, a Nussbaum function-based low-complexity control scheme is designed in [28]. As regards asymptotic tracking control for systems with unknown nonlinearities, an universal global low-complexity controller is proposed in [31]. Nevertheless, it is worth mentioning that the global control of a MAS with unknown nonlinearities is still an unsolved problem, since these methods are based on the condition that the desired output for systems are known, but this knowledge cannot be obtained for some agents of a MAS. Moreover, considering the control coefficients of each agent are time-varying functions, these traditional methods will fall into trouble when solving for the consensus control of a MAS with unknown dynamics.

Motived by the above discussion, we investigate the fully distributed control of a MAS with a directed graph, time-varying control coefficients and completely unknown system nonlinearities. The main contributions of this paper are summarized as follows.

(1) To address the time-varying control coefficients of a MAS, a two-order filter is firstly designed for each agent to produce estimates of the signals from the leader, so that an asymmetric Laplace matrix for a directed graph will not be used to design the controller for each agent of the MAS, by which the difficulty of control design is solved.

(2) To address the completely unknown system nonlinearities in MAS, barrier functions are used to propose a fully distributed controller by combining novel filters; barrier functions are well-suited to dealing with the effects of unknown system nonlinearities, such that global results are achieved, for the first time, in a MAS with completely unknown system nonlinearities in this paper.

(3) To guarantee the prescribed tracking performance by the proposed controller, such that the consensus of the controlled MAS is rigorously proved and all the closed signals are globally bounded.

## 2. Problem Statement and Preliminaries

Consider a class of uncertain MAS as follows

$$
\begin{cases}
\dot{x}_{i,m} = g_{i,m}(\overline{x}_{i,m})x_{i,m+1} + f_{i,m}(\overline{x}_{i,m}) + d_{i,m}(t,\overline{x}_{i,m}), \ m = 1,2\ldots,n-1 \\
\dot{x}_{i,n} = g_{i,n}(\overline{x}_{i,n})u_i + f_{i,n}(\overline{x}_{i,n}) + d_{i,n}(t,\overline{x}_{i,n}) \\
y_i = x_{i,1}, \ for \ i = 1,2,\ldots,N
\end{cases} \tag{1}
$$

where $\overline{x}_{i,m} = [x_{i,1}, x_{i,2}, \ldots, x_{i,m}]^T \in R^m$, $y \in R$, $u \in R$, are the states, the control input and the output of the $i$ th subsystem, respectively. The system nonlinearities $f_{i,m}(\cdot), g_{i,m}(\cdot) : R^m \times R^+ \to R$ are unknown continuous functions with respect to $\overline{x}_{i,m}$. $d_{i,m}(t, \overline{x}_{i,m}), m = 1,2\ldots,n$ represent the system uncertainties and external disturbances.

The desired trajectory for the outputs of the subsystems $y_d$ is bounded and only known by part of the $N$ subsystems, with $\dot{y}_d$ being bounded and unknown to all subsystems.

Suppose that the information transmission condition among the group of $N$ subsystems can be represented by a directed graph $G \triangleq (V, E)$, where $V = \{1, \ldots, N\}$ denotes the set of indexes corresponding to each subsystem. The edge $(i, j) \in E$ indicates that subsystem $j$ could obtain information from subsystem $i$, but not necessarily vice versa. In this case, subsystem $j$ is called a neighbor of subsystem $i$, and vice versa. Denoting the set of neighbors for subsystem $i$ as $N_i \triangleq \{j \in V : (j, i) \in E\}$. Self-edging $(i, i)$ is not allowed, thus $(i, i) \notin E$ and $i \notin N_i$. The connectivity matrix $A = [a_{ij}] \in R^{N \times N}$ of $G$ is defined as $a_{ij} = 1$ if $(j, i) \in E$ and $a_{ij} = 0$ if $(j, i) \notin E$. An in-degree matrix $\Delta$ is introduced, such that $\Delta = diag(\Delta_i) \in R^{N \times N}$ with $\Delta_i = \sum_{j \in N_i} a_{ij}$ being the $i$ th row sum of $A$. Then, the Laplacian matrix of $L$ is defined as $L = \Delta - A$. Defining $B = diag\{\mu_1, \mu_2, \ldots, \mu_N\}$, where $\mu_i = 1$ means the $y_d$ is accessible directly by subsystem $i$, and otherwise, we have $\mu_i = 0$. Throughout this paper, the following notations are used. $\|\cdot\|$ is the Euclidean norm of a vector. Letting $a \in R^n$ and $b \in R^n$ be two vectors, then define the vector operator $.*$ as $a.*b = [a(1)b(1), \ldots, a(n)b(n)]^T$. Letting $Q$ be a matrix, $\lambda_{\min}(Q)$ then denotes the minimum eigenvalue of $Q$.

**Assumption 1.** *The directed graph G contains a spanning tree, and the desired trajectory $y_d$ is accessible to at least one subsystem, i.e., $\sum_{i=1}^{N} \mu_i > 0$.*

**Assumption 2.** *There exist unknown local Lipschitz functions $b_{i,m}(\overline{x}_{i,m})$ such that, for $i = 1,2,\ldots,N$*

$$
|d_{i,m}(t, \overline{x}_{i,m})| \leq b_{i,m}(\overline{x}_{i,m}), m = 1,2,\ldots,n \tag{2}
$$

**Assumption 3.** *The unknown control coefficients $g_{i,m}(\overline{x}_{i,m})$ is strictly positive or negative. Without a loss of generality, it is assumed to be strictly positive, namely, for $i = 1,2,\ldots,N$*

$$
g_{i,m}(\overline{x}_{i,m}) > 0, m = 1,2,\ldots,n \tag{3}
$$

**Lemma 1.** *(Ref. [17]) Based on Assumption 1, the matrix $(L + B)$ is nonsingular. Defining*

$$
\begin{aligned}
&\theta = [\theta_1, \ldots, \theta_N]^T = (L+B)^{-1}[1, \ldots, 1]^T \\
&P = diag\{P_1, \ldots, P_N\} = diag\left\{\frac{1}{\theta_1}, \ldots, \frac{1}{\theta_N}\right\} \\
&Q = P(L+B) + (L+B)^T P
\end{aligned} \tag{4}
$$

*then $\theta_i > 0$ for $i = 1,2,\ldots,N$ and $Q$ is definitely positive.*

**Remark 1.** *In contrast to the methods in [13–16] for a MAS with a directed graph, the control coefficients, $g_{i,m}(\overline{x}_{i,m})$, are time-varying and unknown continuous functions in this paper, which makes the control design much more difficult, since the matrix P in (4) is always unknown and required to be estimated adaptively while the unknown control coefficients $g_{i,1}(\overline{x}_{i,m})$ make P inestimable. To cope with this problem, a novel two-order filter will be given for each agent (shown later).*

**Remark 2.** *The system nonlinearities, $f_{i,m}(\overline{x}_{i,m})$ and $g_{i,m}(\overline{x}_{i,m})$, are completely unknown functions so that there is little knowledge with which to construct the controller. To deal with this problem, neural networks and fuzzy logic systems have been used to approximate the unknown functions caused by the system nonlinearities $f_{i,m}(\overline{x}_{i,m})$ and $g_{i,m}(\overline{x}_{i,m})$ in [22–24], however, only semi-global results can be obtained by use of these approximators. To construct a distributed controller for a MAS with these unknown system nonlinearities with global consensus is a challenging problem, which is solved by the skillfull cooperation of novel two-order filters and barrier functions in the following.*

## 3. Design of Distributed Controller and Filters

In this section, a distributed asymptotic tracking controller for a multi-agent system (1) will be designed. To facilitate the control design in distributed manner, design a filter $(q_{i,1}, q_{i,2})$ for each agent $i$, with $i = 1, \ldots, N$.

### 3.1. Filters Design

Denote

$$z_{i,j} = \sum_{k=1}^{N} a_{i,k}(q_{i,j} - q_{k,j}) + \mu_i(q_{i,j} - y_d^{(j-1)}), j = 1, 2 \tag{5}$$

Then, design the filters as

$$\begin{cases} \dot{q}_{i,1} = q_{i,2} \\ \dot{q}_{i,2} = v_i \end{cases} \tag{6}$$

with

$$v_i = -c_1 \underline{z}_i - c_0 q_{i,2} - c_0 \mathrm{sgn}(\underline{z}_i) \sum_{j=1}^{2} \hat{F}_{i,j} \tag{7}$$

$$\dot{\hat{F}}_{i,j} = \sum_{k=1}^{N} a_{i,k}(\hat{F}_{k,j} - \hat{F}_{i,j}) + \mu_i(F_j - y_d^{(j-1)}), \ j = 1, 2 \tag{8}$$

where $\underline{z}_i = c_0 z_{i,1} + z_{i,2}$, $y_d^{(0)} = y_d$ and $y_d^{(1)} = \dot{y}_d$, and $c_0, c_1$ are design parameters chosen as $c_0 \geq 1$ and $c_1 > c_0 + 1$. We then have the following lemma.

**Lemma 2.** *Consider a closed-loop system consisting of Nfilters (6) satisfying Assumption 1 with local controller (7). The asymptotic consensus tracking of all the filter's outputs to $y_d(t)$ is achieved, i.e., $\lim\limits_{t \to +\infty} |q_{i,1} - y_d(t)| = 0$. Moreover, $q_{i,1}$ and $q_{i,2}$ are bounded.*

**Proof (of Lemma 2).** Consider the following Lyapunov function

$$V_z = \frac{1}{2} z^T P z + \frac{1}{2\gamma} \sum_{j=1}^{2} \widetilde{F}_j^T P \widetilde{F}_j \tag{9}$$

where $z = [\underline{z}_1, \underline{z}_2, \ldots, \underline{z}_N]^T$, $\widetilde{F}_j = \hat{F}_j - F_j$, $\hat{F}_j = [\hat{F}_{1,j}, \hat{F}_{2,j}, \ldots, \hat{F}_{N,j}]^T$, $F_j = [F_{1,j}, F_{2,j}, \ldots, F_{N,j}]^T$, and $\gamma > 0$ is a constant satisfying $\gamma < \frac{2\lambda_{\min}^2(Q)}{\varphi^2}$ with $\varphi = \|P(L + B)\|$. Denote $z_j = [z_{1,j}, z_{2,j}, \ldots, z_{N,j}]^T$ and $q_j = [q_{1,j}, q_{2,j}, \ldots, q_{N,j}]^T$. Then, we have

$$\dot{z} = (L + B)(c_0 q_2 - c_0 y_d^{(1)} + v - y_d^{(2)}) \tag{10}$$

Using (9) and (10), the time derivative of $V_z$ is

$$
\begin{aligned}
\dot{V}_z &= z^T P(L+B)\left(-c_1 z - c_0 \sum_{j=1}^{2} \mathrm{sgn}(\underline{z}).*\underline{F}_j \right. \\
&\quad \left. + c_0 \sum_{j=1}^{2} \varepsilon(\underline{z}).*\underline{F}_j - c_0 y_d^{(1)} - y_d^{(2)}\right) \\
&\quad - \frac{1}{\gamma}\sum_{j=1}^{2} \widetilde{F}_j^T P(L+B)\widetilde{F}_j \\
&\leq -c_1 z^T Q z - \frac{1}{\gamma}\sum_{j=1}^{2}\widetilde{F}_j^T Q \widetilde{F}_j - c_0\sum_{j=1}^{2} z^T P \Delta \mathrm{sgn}(\underline{z}).*\underline{F}_j \\
&\quad + c_0 \sum_{j=1}^{2} z^T P A \mathrm{sgn}(\underline{z}).*\underline{F}_j - c_0\sum_{j=1}^{2} z^T P B \mathrm{sgn}(\underline{z}).*\underline{F}_j \\
&\quad - \sum_{j=1}^{2} z^T P(L+B)(c_0 y_d^{(1)} + y_d^{(2)}) + c_0\sum_{j=1}^{2}\|z\|\|P(L+B)\|\|\widetilde{F}_j\|
\end{aligned}
\tag{11}
$$

where $\mathrm{sgn}(\underline{z}) = [\mathrm{sgn}(\underline{z}_1),\ldots,\mathrm{sgn}(\underline{z}_N)]^T$.

By noting

$$
c_0 \sum_{j=1}^{2} z^T P \Delta \mathrm{sgn}(\underline{z}).*\underline{F}_j = c_0 \sum_{j=1}^{2} F_j \sum_{i=1}^{N} p_i a_{ik}|\underline{z}_i|
\tag{12}
$$

$$
c_0 \sum_{j=1}^{2} z^T P A \mathrm{sgn}(\underline{z}).*\underline{F}_j \leq c_0 \sum_{j=1}^{2} F_j \sum_{i=1}^{N} p_i a_{ik}|\underline{z}_i|
\tag{13}
$$

$$
c_0 \sum_{j=1}^{2} z^T P B \mathrm{sgn}(\underline{z}).*\underline{F}_j = c_0 \sum_{j=1}^{2} F_j \sum_{i=1}^{N} \mu_i p_i|\underline{z}_i|
\tag{14}
$$

$$
\left|\sum_{j=1}^{2} z^T P(L+B)(c_0 y_d^{(1)} + y_d^{(2)})\right| \leq c_0 \sum_{j=1}^{2} F_j \sum_{i=1}^{N} \mu_i p_i|\underline{z}_i|
\tag{15}
$$

$$
\sum_{j=1}^{2} \|z\|\|P(L+B)\|\|\widetilde{\underline{F}}_j\| \leq \lambda_{\min}(Q)\|z\|^2 + \sum_{j=1}^{2} \frac{\varphi^2}{2\lambda_{\min}(Q)}\|\widetilde{\underline{F}}_j\|
\tag{16}
$$

we have

$$
\dot{V}_z \leq -c_2\|z\|^2 - \gamma^*\|\widetilde{\underline{F}}_j\|
\tag{17}
$$

where $c_2 = \lambda_{\min}(Q)(c_1 - c_0)$, $\gamma^* = \lambda_{\min}(Q)\left(\frac{1}{\gamma} - \frac{\varphi^2}{2\lambda_{\min}^2(Q)}\right)$. It is easily verified that $c_2 > 0$ and $\gamma^* > 0$, therefore, it follows from (17) that $\lim\limits_{t \to +\infty}\|z\| = 0$ and hence $\lim\limits_{t \to +\infty}|q_{i,1} - y_d(t)| = 0$. From the boundedness of $V_z$ and $\|z\|$, the boundedness of $q_{i,1}$ and $q_{i,2}$ are easily obtained. This completes the proof.

**Remark 3.** *As is seen, a two-order filter is designed to produce a signal $q_{i,1}$ for each agent. Actually, $q_{i,1}$ is the estimate of $y_d$, as seen in Lemma 2, and the agents no longer require estimating the matrix P. Cooperating these two-order filters makes the use of traditional adaptive control techniques for MAS be easy, and thus the unknown time-varying control coefficients for a MAS with a directed graph can be dealt with.*

*3.2. Design of the Distributed Controller*

In this section, cooperating with the filter (6), the distributed adaptive controller is designed. The following error variables and change of coordinates are introduced

$$
e_{i,1} = \frac{1}{k_i(t)}\left(x_{i,1} - q_{i,1} - \sigma(t)(x_{i,1}^0 - q_{i,1}^0)\right)
\tag{18}
$$

$$
e_{i,m} = \frac{1}{k_i(t)}\left(x_{i,m} - \alpha_{i,m-1} - \sigma(t)x_{i,m}^0\right), m = 2,\ldots,n
\tag{19}
$$

with

$$\sigma(t) = \begin{cases} \frac{1}{t_s^2}(t - t_s)^2, & t < t_s \\ 0, & t \geq t_s \end{cases} \tag{20}$$

where $x_{i,j}^0 = x_{i,j}(0), j = 1, \ldots, n$, and $q_{i,1}^0 = q_{i,1}(0)$, and $t_s$ can be any positive constant. Let $t_s = 1$ in this paper.

Then, the intermediate control signals $\alpha_{i,m}$ and the distributed controller $u_i$ are determined as follows

$$\alpha_{i,m} = -\lambda_{i,m} \frac{e_{i,m}}{1 - e_{i,m}^2}, \quad m = 1, \ldots, n-1 \tag{21}$$

$$u_i = -\lambda_{i,n} \frac{e_{i,n}}{1 - e_{i,n}^2} \tag{22}$$

where $\lambda_{i,m}, 1 \leq i \leq N, 1 \leq m \leq n$ are the positive design parameters. It is easy to verify that $e_{i,m}(0) = 0$ for all $1 \leq i \leq N, 1 \leq m \leq n$ and $e_{i,1}(t) = x_{i,1}(t) - q_{i,1}(t)$ for $t \geq t_s, 1 \leq i \leq N$. $k_i(t)$ are the constrained functions chosen by the designer and used as prescriptive performance functions, satisfying $0 < \underline{k} \leq k_i(t) \leq \overline{k}, |\dot{k}_i(t)| \leq k'$ with $\underline{k}, \overline{k}$ and $k'$ being positive constants.

**Remark 4.** *Function $\sigma(t)$ is constructed to attenuate the influence of the initial conditions, since it makes $e_{i,m}(0) = 0$ and therefore stable results can be achieved under all initial conditions using $\sigma(t)$ for transformation (20). It should also be noted that $\sigma(t)$ of (20) is continuously differentiable and $\dot{\sigma}(t)$ does not exist in the further design of the controller, which means that the designed intermediate control signals and actual controller are smooth.*

### 4. Stability Analysis

In this section, we will give the main results with the designed fully distributed controller and present the stability analysis. The main results of this article are as follows.

**Theorem 1.** *Consider the closed-loop system consisting of N uncertain agents as (1) satisfying Assumptions 1–3, the intermediate control signals (21) and the distributed controller (22). Then, we have the following properties:*

(1)  *All the signals in the closed-loop system are globally bounded*
(2)  *Prespecified tracking performance can be guaranteed, namely, $|e_{i,1}| < 1$, for $i = 1, 2, \ldots, N$.*
(3)  *The output of each agent ultimately satisfies $|y_i - y_d| \leq k_i(t)$.*

**Proof (of Theorem 1).** From (18), (19) and (21), we have

$$x_{i,1} = k_i e_{i,1} + q_{i,1} + \sigma(t)(x_{i,1}^0 - q_{i,1}^0) \tag{23}$$

$$x_{i,m} = k_i e_{i,m} + \alpha_{i,m-1}(e_{i,m-1}) + \sigma(t)x_{i,m}^0, \quad m = 2, \ldots, n \tag{24}$$

It can be observed from (23) that $x_{i,1}$ is continuous function of $e_{i,1}, q_{i,1}$ and $\sigma(t)$, where $q_{i,1}$ and $\sigma(t)$ are bounded time-varying functions. Thus, $x_{i,1}$ can be rewritten as the form of continuous function of $e_{i,1}$ and $t$. Similar analysis can be made for $x_{i,m}$. Therefore, we obtain

$$\begin{aligned} \dot{e}_{i,1} &= \frac{1}{k_i}(f_{i,1}(x_{i,1}) + g_{i,1}(x_{i,1})x_{i,2} - q_{i,2} - \dot{\sigma}(t)(x_{i,1}^0 - q_{i,1}^0) - \dot{k}_i e_{i,1} + d_{i,1}(t, x_{i,1}) \\ &= h_{i,1}(t, e_{i,1}, e_{i,2}, \hat{v}_i) \end{aligned} \tag{25}$$

$$\begin{aligned} \dot{e}_{i,m} &= \frac{1}{k_i}(f_{i,m}(\overline{x}_{i,m}) + g_{i,m}(\overline{x}_{i,m})x_{i,m+1} - \frac{\partial \alpha_{i,m-1}}{\partial e_{i,m-1}} h_{i,m-1}(t, e_{i,1}, \ldots, e_{i,m}) \\ &\quad - \dot{\sigma}(t)x_{i,m}^0 - \dot{k}_i e_{i,m} + d_{i,m}(t, \overline{x}_{i,m})) \\ &= h_{i,m}(t, e_{i,1}, \ldots, e_{i,m+1}) \end{aligned} \tag{26}$$

where $e_{i,n+1} = 0$, and $h_{i,m}(\cdot)$, $m = 1, 2, \ldots, n$ are some continuous functions. Defining $e_i = [e_{i,1}, \ldots, e_{i,n}]^T$ and in view of (25) and (26), we obtain

$$
\dot{e}_i = h_i(t, e_i) = \begin{bmatrix} h_{i,1}(t, e_{i,1}) \\ h_{i,2}(t, e_{i,1}, e_{i,2}) \\ \vdots \\ h_{i,n}(t, e_{i,1}, \ldots, e_{i,n}) \end{bmatrix} \tag{27}
$$

Let us define the open set:

$$
\Omega_e = \underbrace{(-1, 1) \times \cdots \times (-1, 1)}_{n-times} \tag{28}
$$

It is easily observed that $e_i(0) \in \Omega_e, i = 1, 2, \ldots, N$. Additionally, $h_{i,m}(\cdot)$, $m = 1, 2, \ldots, n$ are continuous with respect to all its variables, owing to the fact that $y_d, \dot{y}_d, \sigma(t), q_{i,1}, k_i(t), f_{i,m}, g_{i,m}, \alpha_{i,m}$ are all continuous differentiable functions. Therefore, it follows from [32] that the conditions on $h_{i,m}(\cdot)$ ensure the existence and uniqueness of a maximal solution $\eta_i(t)$ on the time interval $[0, t_{\max})$, namely, $e_i(t) \in \Omega_e$ for $t \in [0, t_{\max})$, which implies

$$
e_{i,m} \in (-1, 1), for \ \forall t \in [0, t_{\max}) \tag{29}
$$

for $i = 1, \ldots, N$, and $m = 1, \ldots, n$.

In the following, we will prove that $t_{\max} = +\infty$ by seeking a contradiction. Suppose that $t_{\max} < +\infty$; then the related analysis is performed as follows, and all of what follows is based on $t \in [0, t_{\max})$.

Step 1: Consider the following positive definite functions

$$
V_{i,1} = \frac{1}{2} \log \frac{1}{1 - e_{i,1}^2} \tag{30}
$$

Let $\xi_{i,1} = \frac{1}{1 - e_{i,1}^2}$. It follows from (21), (24), (25) and (30) that the time derivative of $V_{i,1}$ is

$$
\begin{aligned}
\dot{V}_{i,1} &= \frac{\xi_{i,1}}{k_i}(f_{i,1}(x_{i,1}) + g_{i,1}(x_{i,1})(\alpha_{i,1} + k_i e_{i,2} + \sigma(t) x_{i,2}^0) + \\
&\quad - q_{i,2} - \dot{\sigma}(t)(x_{i,1}^0 - q_{i,1}^0) - \dot{k}_i e_{i,1} + d_{i,1}(t, x_{i,1})) \\
&\leq \frac{1}{k_i} g_{i,1}(x_{i,1}) \alpha_{i,1} \xi_{i,1} + F_{i,1}(t) |\xi_{i,1}| \\
&\leq -\lambda_{i,1} E_{i,1} \xi_{i,1}^2 + F_{i,1} |\xi_{i,1}|
\end{aligned} \tag{31}
$$

where $F_{i,1} = \frac{1}{k_i}(|f_{i,1}(x_{i,1})| + |g_{i,1}(x_{i,1})| \left| k_i e_{i,2} + \sigma(t) x_{i,2}^0 \right| + |q_{i,2}| + \left| \dot{\sigma}(t)(x_{i,1}^0 - q_{i,1}^0) \right| + \left| \dot{k}_i e_{i,1} \right| + b_{i,1}(x_{i,1}))$ and $E_{i,1} = \frac{g_{i,1}(x_{i,1})}{k_i}$. Note that $x_{i,1}$, $e_{i,1}$ and $e_{i,2}$ are bounded on $\Omega_e$ because (23) and (29), respectively. Utilizing the fact that $k_i(t), \dot{k}_i(t), \sigma(t), q_{i,1}, q_{i,2}$ are bounded and employing the extreme value theorem, owing to the continuity of $f_{i,1}(\cdot), g_{i,1}(\cdot)$ and $b_{i,1}(\cdot)$, we arrive at

$$
E_{i,1} \geq c_{1,1} > 0 \tag{32}
$$

$$
c_{3,1} \geq F_{i,1} \geq c_{2,1} \geq 0 \tag{33}
$$

where $c_{1,1}, c_{2,1}$, and $c_{3,1}$ are some unknown positive constants.

Then, substituting (32) and (33) into (31) yields

$$
\dot{V}_{i,1} \leq -\lambda_{i,1} c_{1,1} \xi_{i,1}^2 + c_{3,1} |\xi_{i,1}| \tag{34}
$$

From (34), it follows that $\dot{V}_{i,1}$ is negative when $|\xi_{i,1}| \leq c_{3,1}/\lambda_{i,1} c_{1,1}$ and subsequently that

$$
|\xi_{i,1}| \leq \xi_{i,1}^* = \frac{c_{3,1}}{\lambda_{i,1} c_{1,1}} \tag{35}
$$

which implies

$$|e_{i,1}(t)| \leq c_{4,1} = 1 - \frac{1}{\xi_{i,1}^{*2}} < 1 \tag{36}$$

As a result, the control signal $\alpha_{i,1}$ is bounded. Moreover, invoking (24), we also can conclude the boundedness of $x_{i,2}$. Therefore, the time derivative of $\alpha_{i,1}$ is

$$\dot{\alpha}_{i,1} = -\lambda_{i,1}\dot{\xi}_{i,1} \tag{37}$$

where

$$
\begin{aligned}
\left|\dot{\xi}_{i,1}\right| &\leq \frac{(1+e_{i,1}^2)}{k_i(1-e_{i,1}^2)^2}\left(|f_{i,1}(x_{i,1})| + \left|g_{i,1}(x_{i,1})(k_i e_{i,2} + \rho(t)x_{i,2}^0 + \alpha_{i,1})\right|\right. \\
&\left.+|q_{i,2}| + \left|\dot{\sigma}(t)(x_{i,1}^0 - q_{i,1}^0)\right| + \left|\dot{k}_i e_{i,1}\right| + b_{i,1}(x_{i,1})\right)
\end{aligned}
\tag{38}
$$

Noting (36) and using the same analysis as (33), it also easy to conclude the boundedness of $\dot{\xi}_{i,1}$, and hence $\dot{\alpha}_{i,1}$.

Step $j$ ($2 \leq j \leq n$): Consider the following positive definite functions

$$V_{i,j} = \frac{1}{2}\log\frac{1}{1 - e_{i,j}^2} \tag{39}$$

Let $\xi_{i,j} = \frac{1}{1 - e_{i,j}^2}$. In a similar fashion to that in the former step, by noting Assumption 1, it follows from (21), (24), (26) and (39) that the time derivative of $V_{i,j}$ is

$$
\begin{aligned}
\dot{V}_{i,j} &= \frac{\xi_{i,j}}{k_i}\left(f_{i,j}(\overline{x}_{i,j}) + g_{i,j}(\overline{x}_{i,j})(\alpha_{i,j} + k_i e_{i,j+1} + \sigma(t)x_{i,j+1}^0)\right. \\
&\left. \quad -\dot{\alpha}_{i,j-1} - \dot{\sigma}(t)x_{i,j}^0 - \dot{k}_i e_{i,j} + d_{i,j}(t,\overline{x}_{i,j})\right) \\
&\leq \frac{1}{k_i}g_{i,j}(\overline{x}_{i,j})\alpha_{i,j}\xi_{i,j} + F_{i,j}|\xi_{i,j}| \\
&\leq -\lambda_{i,j}E_{i,j}\xi_{i,j}^2 + F_{i,j}|\xi_{i,j}|
\end{aligned}
\tag{40}
$$

where $F_{i,j} = \frac{1}{k_i}\left(|f_{i,1}(\overline{x}_{i,j})| + |g_{i,1}(\overline{x}_{i,j})|\left|k_i e_{i,j+1} + \sigma(t)x_{i,j+1}^0\right| + |\dot{\alpha}_{i,j-1}| + \left|\dot{\sigma}(t)x_{i,j}^0\right| + \left|\dot{k}_i e_{i,j}\right| + b_{i,j}(\overline{x}_{i,j})\right)$ and $E_{i,j} = \frac{\pi g_{i,j}(x_{i,j})}{2k_i}$. Noting that $x_{i,m}, m = 1,2,\ldots,j$ are bounded on $\Omega_e$ because the boundedness of $\alpha_{i,m-1}$, $e_{i,j}$ and $e_{i,j+1}$ are bounded on $\Omega_e$ in view of (29). Utilizing the fact that $k_i(t)$, $\dot{k}_i(t)$ are bounded and employing the extreme value theorem owing to the continuity of $f_{i,j}(\cdot)$, $g_{i,j}(\cdot)$ and $b_{i,j}(\cdot)$, we arrive at

$$E_{i,j} \geq c_{1,j} > 0 \tag{41}$$

$$c_{3,j} \geq F_{i,j} \geq c_{2,j} \geq 0 \tag{42}$$

with $c_{1,j}$, $c_{2,j}$ and $c_{3,j}$ being some unknown positive constants.

Then, substituting (41) and (42) into (40) yields

$$\dot{V}_{i,j} \leq -\lambda_{i,j}c_{1,j}\xi_{i,j}^2 + c_{3,j}|\xi_{i,j}| \tag{43}$$

From (43), it follows that $\dot{V}_{i,j}$ is negative when $|\xi_{i,j}| \leq c_{3,j}/\lambda_{i,j}c_{1,j}$ and subsequently that

$$|\xi_{i,j}| \leq \xi_{i,j}^* = \frac{c_{3,j}}{\lambda_{i,j}c_{1,j}} \tag{44}$$

which implies

$$|e_{i,j}(t)| \leq c_{4,j} = 1 - \frac{1}{\xi_{i,j}^{*2}} < 1 \tag{45}$$

As a result, the control signal $\alpha_{i,j}$ is bounded. Moreover, we also can conclude the boundedness of $x_{i,j+1}$ by noting (24). Finally, the time derivative of $\alpha_{i,j}$ is

$$\dot{\alpha}_{i,j} = -\lambda_{i,j}\dot{\xi}_{i,j} \tag{46}$$

where

$$
\begin{aligned}
\left|\dot{\xi}_{i,j}\right| &\leq \frac{(1+e_{i,j}^2)}{k_i(1-e_{i,j}^2)^2}\left(\left|f_{i,j}(\overline{x}_{i,j})\right| + \left|g_{i,j}(\overline{x}_{i,j})(k_ie_{i,j+1} + \sigma(t)x_{i,j+1}^0 + \alpha_{i,j})\right| \right. \\
&\quad + \left|\dot{\sigma}(t)x_{i,j}^0\right| + \left|\dot{k}_ie_{i,j}\right| + b_{i,j}(\overline{x}_{i,j}))
\end{aligned} \tag{47}
$$

Noting (45) and using the same analysis as (42), it also easy to conclude the boundedness of $\dot{\xi}_{i,j}$ and hence $\dot{\alpha}_{i,j}$.

Step $n$: Consider the following Lyapunov functions

$$V_{i,n} = \frac{1}{2}\log\frac{1}{1-e_{i,n}^2} \tag{48}$$

Let $\xi_{i,n} = \frac{1}{1-e_{i,n}^2}$. Similar as the former steps, we can have

$$\dot{V}_{i,n} \leq -\lambda_{i,n}c_{1,n}\xi_{i,n}^2 + c_{3,n}|\xi_{i,n}| \tag{49}$$

where $c_{1,n}$ and $c_{3,n}$ are some unknown positive constants. It follows from (49) that $\dot{V}_{i,n}$ is negative when $|\xi_{i,n}| \leq c_{3,n}/\lambda_{i,n}c_{1,n}$ and subsequently that

$$|\xi_{i,n}| \leq \xi_{i,n}^* = \frac{c_{3,n}}{\lambda_{i,n}c_{1,n}} \tag{50}$$

which implies

$$|e_{i,n}(t)| \leq c_{4,n} = 1 - \frac{1}{\xi_{i,n}^{*2}} < 1 \tag{51}$$

As a result, the control signal $\alpha_{i,j}$ is bounded. Moreover, we also can conclude the boundedness of $u_i$. Notice that (36), (45) and (51) imply that $e_i(t) \in \Omega'_e$, for $\forall t \in [0, t_{\max}), i = 1, 2, \ldots, N$, where the set $\Omega'_e$ is nonempty and compact, defined as

$$\Omega'_e = [-c_{4,1}, c_{4,1}] \times [-c_{4,2}, c_{4,2}] \cdots \times [-c_{4,n}, c_{4,n}]$$

Owing to (36), (45) and (51) it is straightforward to verify that $\Omega'_e \subset \Omega_e$. Therefore, assuming $t_{\max} < +\infty$ dictates the existence of a time instant $t' \in [0, t_{\max})$, such that $e_i(t') \notin \Omega'_e$, which is a clear contradiction. Therefore, $t_{\max} = +\infty$. Hence, all closed-loop signals remain bounded and moreover $e_i(t) \in \Omega'_e \subset \Omega_e$, *for* $\forall t \geq 0$. Furthermore, from (36) we conclude that

$$|e_{i,1}(t)| \leq c_{4,1} < 1 \tag{52}$$

Then, for all $t \geq 0$. In view of Lemma 2 and (52), we have

$$
\begin{aligned}
\lim_{t\to+\infty}|y_i - y_d| &= \lim_{t\to+\infty}|q_{i,1} - y_d + y_i - q_{i,1}| \\
&\leq \lim_{t\to+\infty}|q_{i,1} - y_d| + \lim_{t\to+\infty}|y_i - q_{i,1}| \\
&\leq \lim_{t\to+\infty}|k_ie_{i,1}(t)| \\
&\leq k_i
\end{aligned} \tag{53}
$$

This completes the proof.

## 5. Simulation Study

Two examples will be given to demonstrate the effectiveness of the proposed distributed adaptive controller in this section, as follows.

**Example 1.** *Consider the following multi-agent systems*

$$
\begin{cases}
\dot{x}_{i,1} = g_{i,1}(\overline{x}_{i,1})x_{i,2} + f_{i,1}(t, \overline{x}_{i,1}), \\
\dot{x}_{i,2} = g_{i,2}(\overline{x}_{i,2})u_i + f_{i,2}(t, \overline{x}_{i,2}) \\
y_i = x_{i,1}, \ for \ i = 1, 2, 3, 4
\end{cases}
$$

*with the system functions chosen as follows:* $f_{1,1} = x_{1,1}^2$, $g_{1,1} = 1 + x_{1,1}^2$, $f_{1,2} = x_{1,1}x_{1,2}$, $g_{1,2} = 1$, $f_{2,1} = x_{2,1}^3 + 0.2\sin t$, $g_{2,1} = 1 + 0.1\cos x_{2,1}$, $f_{2,2} = x_{2,1}x_{2,2}$, $g_{2,2} = 1$, $f_{3,1} = x_{3,1}\sin x_{3,1}$, $g_{3,1} = 1$, $f_{3,2} = x_{3,1}x_{3,2} + 0.1\sin t$, $g_{3,2} = 1$, $f_{4,1} = x_{4,1} + 0.8 + 0.2\sin t$, $g_{4,1} = 1$, $f_{4,2} = x_{4,1}x_{4,2} + 0.2\cos t$, $g_{4,2} = 1$. *The communication topology for these subsystems are depicted in* Figure 1.

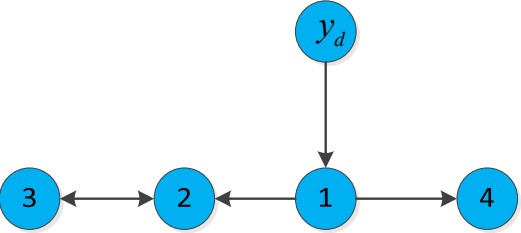

**Figure 1.** Communication topology for four subsystems.

The desired trajectory for the outputs of each subsystem is $y_d = \sin t$. The initial conditions for each subsystems are set as: $x_{1,1}(0) = 0.5$, $x_{2,1}(0) = -0.5$, $x_{3,1}(0) = 0$, $x_{4,1}(0) = 0.1$ and $x_{1,2}(0) = x_{2,2}(0) = x_{3,2}(0) = x_{4,2}(0) = 0$. Then, the intermediate control signals are designed and the distributed controllers are designed as follows

$$
\alpha_{i,1} = -\lambda_{i,1}\frac{e_{i,1}}{1 - e_{i,1}^2}, \ i = 1, 2, 3, 4
$$

$$
u_i = -\lambda_{i,2}\frac{e_{i,2}}{1 - e_{i,2}^2}, \ i = 1, 2, 3, 4
$$

where their control parameters and functions are selected as: $\lambda_{1,1} = 5$, $\lambda_{2,1} = 5$, $\lambda_{3,1} = 5$, $\lambda_{4,1} = 4$, $\lambda_{1,2} = 10$, $\lambda_{2,2} = 20$, $\lambda_{3,2} = 10$ and $\lambda_{4,2} = 10$, $k_i(t) = 3e^{-0.5t} + 0.01$ for $i = 1, 2, 3, 4$. For the filters, the parameters are chosen as: $c_0 = 2$ and $c_1 = 6$. Then, the simulation results are reported as Figures 2–4.

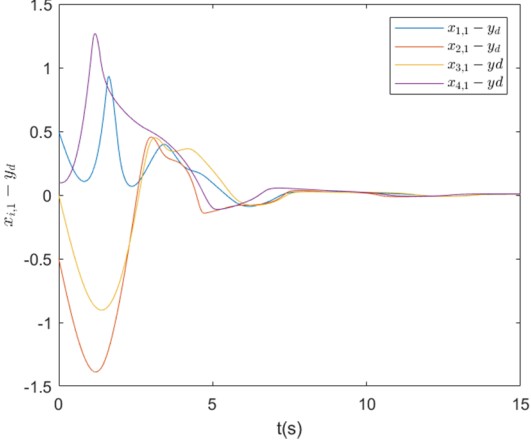

**Figure 2.** Tracking errors $x_{i,1} - y_d$ for $1 \le i \le 4$.

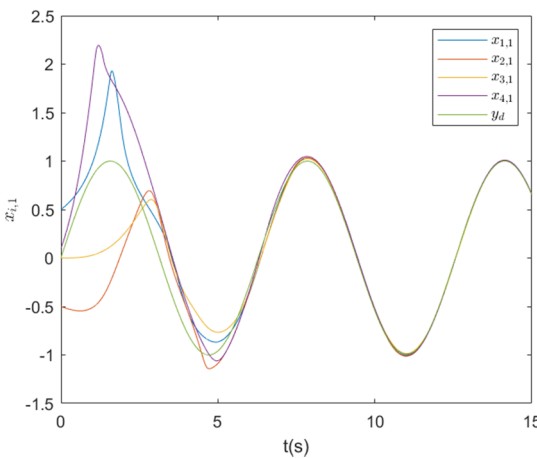

**Figure 3.** Outputs $x_{i,1}$ for $1 \leq i \leq 4$ and $y_d$.

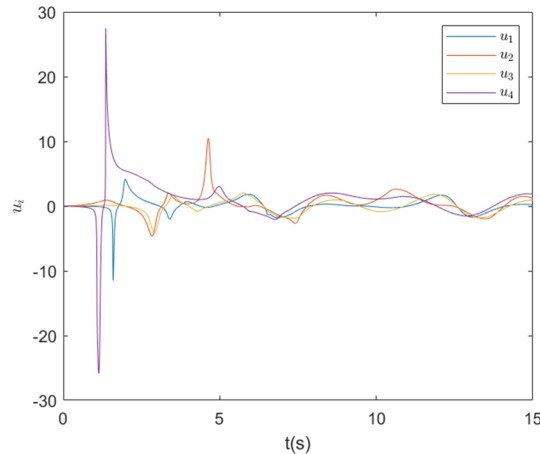

**Figure 4.** Distributed control inputs $u_i$ for $1 \leq i \leq 4$.

It can be observed from Figures 2–4 that under the designed distributed controllers, the outputs of the subsystems track the desired trajectory very quick, and the tracking performance is satisfactory.

**Example 2.** *Consider the consensus for four high-maneuver fighters, with communication topologies as in Figure 5 and their flight control systems as follows [33].*

$$\begin{cases} \dot{X}_{i,1} = f_1(X_{i,1}, X_{i,3}) + G_1(X_{i,1})X_{i,2} \\ \dot{X}_{i,2} = f_2(X_i) + G_2 u_i \\ \dot{X}_{i,3} = f_3(X_i) \end{cases} \tag{54}$$

*with*

$$f_1(X_{i,1}, X_{i,3}) = \begin{bmatrix} q_i \tan \theta_i \sin \phi_i + r_i \tan \theta_i \cos \phi_i \\ p_i \beta_i + z_0 \Delta \alpha_i + (g_0/V_i)(\cos \theta_i \cos \phi_i - \cos \theta_0) \\ y_\beta \beta_i + p_i(\sin \alpha_0 + \Delta \alpha_i) + (g_0/V_i) \cos \theta_i \sin \phi_i \end{bmatrix}$$

$$G_1(X_{i,1}) = \begin{bmatrix} 1 & 0 & 0 \\ 0 & 1 & 0 \\ 0 & 0 & -\cos \alpha_0 \end{bmatrix}$$

$$f_2(X_i) = \begin{bmatrix} l_\beta \beta_i + l_p p_i + l_q q_i + l_r r_i + (l_{\beta\alpha}\beta_i + l_{r\alpha}r_i)\Delta\alpha_i - i_1 q_i r_i \\ m_\alpha \Delta\alpha_i + m_q q_i + i_2 p_i r_i - m_{\dot{\alpha}}(g_0/V_i)(\cos\theta_i \cos\phi_i - \cos\theta_0) \\ n_\beta \beta_i + n_r r_i + n_p p_i + n_{p\alpha} p_i \Delta\alpha_i - i_3 p_i q_i + n_q q_i \end{bmatrix}$$

$$G_2 = [L, M, N]^T$$

$$L = [l_{\delta_{el}}, l_{\delta_{er}}, l_{\delta_{al}}, l_{\delta_{ar}}, 0, 0, l_{\delta_r}]^T$$

$$M = [m_{\delta_{el}}, m_{\delta_{er}}, m_{\delta_{al}}, m_{\delta_{ar}}, m_{\delta_{lef}}, m_{\delta_{tef}}, m_{\delta_r}]^T$$

$$N = [n_{\delta_{el}}, n_{\delta_{er}}, n_{\delta_{al}}, n_{\delta_{ar}}, 0, 0, n_{\delta_r}]^T$$

$$f_3(X_i) = q_i \cos\phi_i - r_i \sin\phi_i$$

where $X_i = \left(X_{i,1}^T, X_{i,2}^T\right)^T = (\phi_i, \alpha_i, \beta_i, p_i, q_i, r_i, \theta_i)^T$ are the roll angle, attack angle, sideslip angle, roll angular velocity, pitching angular velocity, yaw angular velocity and pitch angle of fighter i, respectively. $y_i = X_{i,1} = [\phi_i, \alpha_i, \beta_i]^T$ $X_{i,2} = [p_i, q_i, r_i]^T$ $X_{i,3} = \theta_i$. $u_i = [\delta_{iel}, \delta_{ier}, \delta_{ial}, \delta_{iar}, \delta_{ilef}, \delta_{itef}, \delta_{ir}]^T$ are the left and right elevators, left and right ailerons, front and rear flaps, and rudder, respectively. Detailed explanations for the parameters and variables of this model can be found in [26]. Suppose that they are all flying at an altitude of 40,000 feet, at a speed of 0.6 Mach. The desired output for these fighters is $y_d = [y_{d,1}, y_{d,2}, y_{d,3}]^T = [20, 30, 0]^T$. The signal $y_d$ is only available for fighter 1.

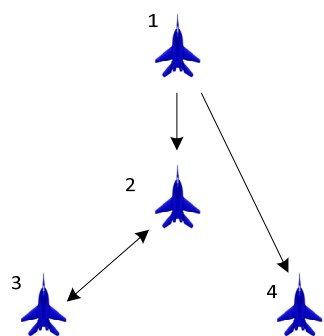

**Figure 5.** Communication topology for four fighters.

According to Theorem 1, we design the distributed flight controller as follows

$$\xi_{i,1} = G_1^{-1}(X_{i,1})diag\left\{-\lambda_{i,1}\frac{e_{i,\phi}}{1-e_{i,\phi}^2}, -\lambda_{i,1}\frac{e_{i,\alpha}}{1-e_{i,\alpha}^2}, -\lambda_{i,1}\frac{e_{i,\beta}}{1-e_{i,\beta}^2}\right\}$$

$$u_i = G_2^+ diag\left\{-\lambda_{i,2}\frac{e_{i,p}}{1-e_{i,p}^2}, -\lambda_{i,2}\frac{e_{i,q}}{1-e_{i,q}^2}, -\lambda_{i,2}\frac{e_{i,r}}{1-e_{i,r}^2}\right\}$$

with

$$e_{i,\phi} = \phi_i - q_{d,1}, e_{i,\alpha} = \alpha_i - q_{d,2}, e_{i,\beta} = \beta_i - q_{d,3}$$
$$[e_{i,p}, e_{i,q}, e_{i,r}]^T = [p_i, q_i, r_i]^T - \xi_{i,1}$$

where $q_{d,1}$, $q_{d,2}$ and $q_{d,3}$ are the signals produced by filter (6) with $y_{d,i}$, $i = 1, 2, 3$ being the filter inputs, respectively. $\lambda_{i,1} = 1$ and $\lambda_{i,2} = 2$ for $i = 1, 2, 3, 4$, and $G_2^+$ represents the pseudo-inverse for $G_2$.

For the purposes of comparison, we use the control method of [17] under the same conditions. Following [17], the controller for the distributed flight controller is designed as follows

$$\xi_{i,1} = G_1^{-1}(X_{i,1})diag\{-\lambda_{i,1}e_{i,\phi}, -\lambda_{i,1}e_{i,\alpha}, -\lambda_{i,1}e_{i,\beta}\}$$
$$u_i = G_2^+ diag\{-\lambda_{i,2}e_{i,p}, -\lambda_{i,2}e_{i,q}, -\lambda_{i,2}e_{i,r}\}$$

where the variables and controller parameters are the same as in our proposed methods. The simulation results are then reported in Figures 6–10. In Figure 6, the dotted curves

denote the outputs of fighters under the control of the method in [17], while the solid curves denote the outputs of fighters under the control of method in this paper. It can be seen from Figure 6 that our control performance is better than [17] since the outputs of ours track the desired value more accurately. Figures 7–10 show the actions of actuators of four fighters under our method. Figure 11 show the controller performance of our method and that from [17]. In Figure 11, the blue curves denote the control efforts $E_1$ of the fighters with our method, while the red curves denote the control efforts $E_2$ of Fighters in the method from [17], where $E_1$ and $E_2$ are defined as

$$E_k = \sqrt{\delta_{iel}^2 + \delta_{ier}^2 + \delta_{ial}^2 + \delta_{iar}^2 + \delta_{ilef}^2 + \delta_{itef}^2 + \delta_{ir}^2}$$
$$k = 1, 2 \; and \; i = 1, 2, 3, 4$$

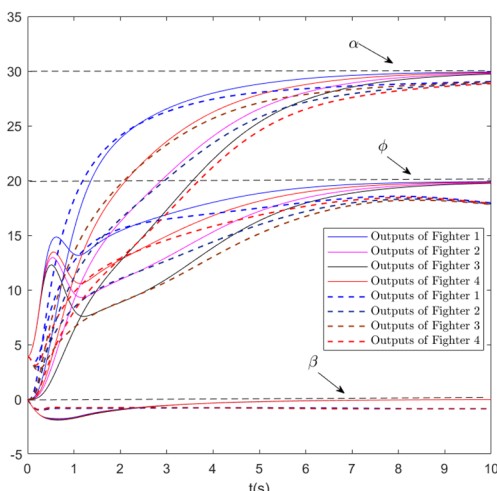

**Figure 6.** Output of four fighters.

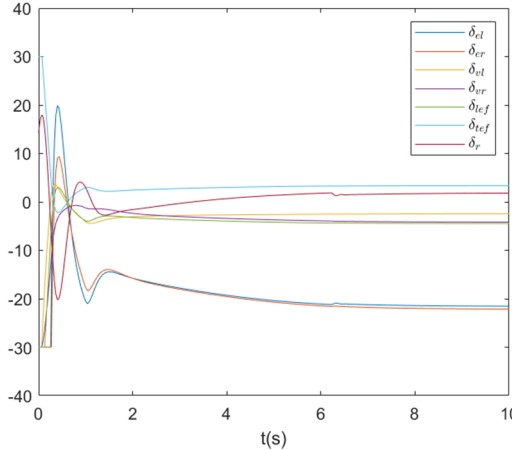

**Figure 7.** Actuator actions of Fighter 1.

It can be seen from Figure 11 that, initially, the control efforts of our method are greater than those in [17], and finally, there is little difference in effort between these methods, which means that the control performance of our method is better under similar control efforts.

It can be seen from these results that the consensus between the four fighters is achieved and the tracking performance is very good, while fairly good control performance is achieved.

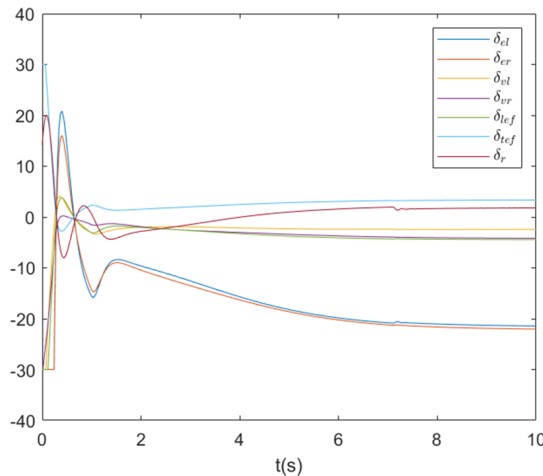

**Figure 8.** Actuator actions of Fighter 2.

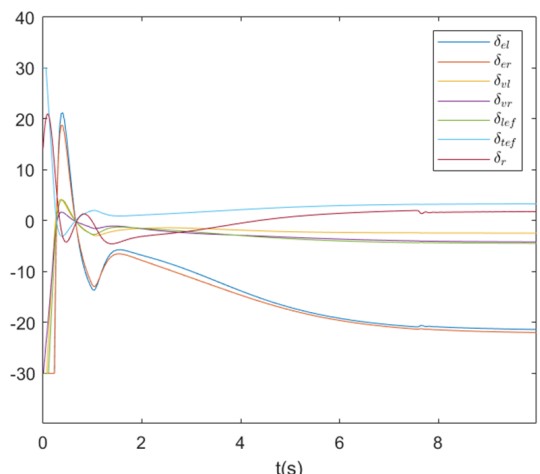

**Figure 9.** Actuator actions of Fighter 3.

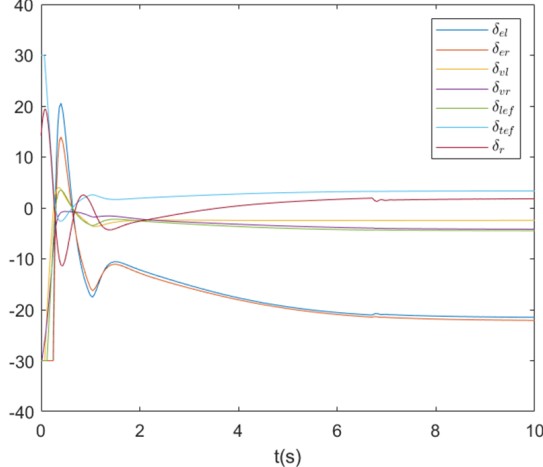

**Figure 10.** Actuator actions of Fighter 4.

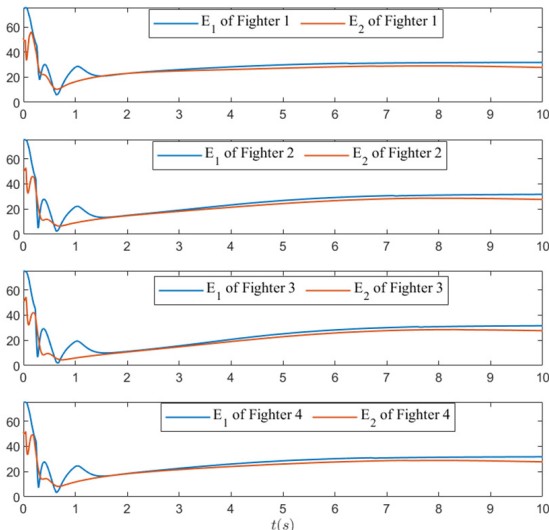

**Figure 11.** Control efforts.

## 6. Conclusions

A novel distributed consensus method was presented for a MAS with completely unknown system nonlinearities and time-varying control coefficients under a directed graph. A two-order filter for each agent was constructed, providing the desired signals and thus avoiding estimating the unknown matrix, which is related on a Laplace matrix. Combined with these filters, a global consensus method was proposed for a MAS with completely unknown system nonlinearities under a directed graph for the first time. The proposed consensus method was applied to two examples. It was shown that four high-maneuver fighters achieved angular consensus and had very good control performances using the proposed method. The two simulation results demonstrated the effectiveness of the proposed method.

**Author Contributions:** Funding acquisition, W.F.; Investigation, H.H.; Writing—original draft, Z.L.; Writing—review and editing, S.L. and Q.L. All authors have read and agreed to the published version of the manuscript.

**Funding:** This research was funded by National Natural Science Foundation of China under grant 62106284 and 62176214, and also in partly funded by Nature Science Foundation of Shaanxi Province of China under grant 2021JQ-370 and also in partly funded by Xi'an Youth Talent Promotion Plan under grant 095920201309.

**Conflicts of Interest:** The authors declare no conflict of interest.

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
