# Peer review of "Fully Distributed Control for a Class of Uncertain Multi-Agent Systems with a Directed Topology and Unknown State-Dependent Control Coefficients"

_applsci, doi:10.3390/app112311304_

Round 1

Reviewer 1 Report

The paper presents an approach for non-linear control for distributed systems based on a consensus algorithm with limited information between the individual subsystems. The results are interesting, however, there are numerous typos and incorrect use of English which makes it very challenging to read.

Figure 1: add two arrows indicating the communication between 2 and 3

Typos:

  • line 33: remove "there are" instead say "Recently, some significant progress has been made"
  • line 44: avoid idioms and say "Though much progress has been made"
  • line 88: remove the extra "is" in "is of G", just say "of G"
  • line 165: remove "always holds"
  • line 167: it should say "continuously differentiable"
  • line 180: no need to say "ultimately"
  • equation (24), add a space before "m"
  • line 186: another missing space before "m"
  • line 187: remove the period and the capital "In"
  • line 194: the comma should be on the previous line
  • line 243: consider putting the set omega on a separate line

Reviewer 2 Report

Modern control systems of linear and nonlinear phenomena based on mathematical modeling of processes are an important part involving control systems theory and applications. Nonetheless, a rigorous knowledge and characterization (quantified) of their nonlinearities, parameters and/or control coefficients are almost always a great challenge.

The paper under review discusses the possibility of implementing a control of uncertain multi-agent systems (MAS) with completely unknown system nonlinearities and unknown control coefficients, and a global consensus method is proposed by constructing novel filters and barrier functions-based distributed controllers.

In this context, the paper is relevant to the field and of particular scientific importance. Furthermore, in general, the paper is well written and brings a contribution to the state-of-art of this research area. However, I suggest some modifications:

- is the novelty presented restricted mainly to an analysis of unknown system nonlinearities and unknown control coefficients ? Maybe it is of reason to clarify it more forcefully;

- the review of the literature is not long enough and still misses some relevant and recent papers;

- more detailed information should be provided, e.g., how the proposed algorithm was implemented to generate Figs. 2-4 and Figs. 6-10 ? Please clarify.

- What is the computational complexity of the proposed controller? Could it be implemented as embedded software? Or what about real-time control ?

- a robust controller performance evaluation is missing. The performance of the controller should be analyzed in the time and frequency domains (ex., by using Root-Mean-Squares (RMS) and Power Spectral Density (PSD) values).

- a performance comparison (including statistical treatment) between the proposed method and existing approaches published in the literature is also very important in order to certify the efficiency of the method proposed by authors in this study.

- a deeper discussion about the achieved results can improve the perception of impact of the proposed manuscript;

- additional proofreading is needed to fix some typos.

Round 2

Reviewer 2 Report

I appreciate the effort made by authors in addressing my observations. The paper has improved and the authors managed to address some of my questions. Nonetheless, important concerns still remain because 2 questions pointed in my previous review are not satisfactorily explained/addressed, namely:
1) Regarding my previous question: - a robust controller performance evaluation is missing. The performance of the controller should be analyzed in the time and frequency domains (ex., by using Root-Mean-Squares (RMS) and Power Spectral Density (PSD) values).
If authors don’t think their controller has better performance than other controllers, so I wonder: What is the sense proposing this controller involving unknown system nonlinearities ? This journal is entitled "Applied Sciences" and I understand (regarding applied sciences) that proposed  new methods or approaches should point to more interesting and/or efficient to provide solutions. Or (at least) they should conclude that the new proposed method (or approach) is not better when compared to the others methods found in the literature. For this reason, some level of performance evaluation capable to provide comparisons between existing apporaches/methods vis-a-vis the present proposed method should be provided. Although the authors state their difficulty to analyze in frequency domain, it is possible to provide some analysis in the time domain. Moreover, a performance comparison (including statistical treatment) is very important.

2) And also my previous question: - a deeper discussion about the achieved results can improve the perception of impact of the proposed manuscript.
The authors included some paragraphs in 'Introduction' section. Nonetheless, a deeper discussion should be provided related to results achieved in this present paper (results obtained by authors of the present study, after simulation results). This deeper discussion remains missing. 

Please check also for other typos.  
